# The Double Face of miR-708: A Pan-Cancer Player with Dissociative Identity Disorder

**DOI:** 10.3390/genes13122375

**Published:** 2022-12-16

**Authors:** Jaqueline Carvalho de Oliveira, Carolina Mathias, Verônica Cristina Oliveira, Julia Alejandra Pezuk, María Sol Brassesco

**Affiliations:** 1Department of Genetics, Federal University of Paraná, Curitiba 80060-000, Brazil; 2Laboratory of Applied Science and Technology in Health, Carlos Chagas Institute, Oswaldo Cruz Foundation (Fiocruz), Curitiba 81350-010, Brazil; 3Department of Biotechnology and Health Innovation, Anhanguera University of São Paulo, Pirituba 05145-200, Brazil; 4Biology Department, Faculty of Philosophy, Sciences and Letters at Ribeirão Preto, University of São Paulo, Ribeirão Preto 14040-901, Brazil

**Keywords:** microRNA, miR-708, hsa-miR-508-5p, hsa-miR-508-3p, cancer

## Abstract

Over the last decades, accumulating evidence has shown tumor-dependent profiles of miR-708, being either up- or downregulated, and thus, acting as a “Janus” regulator of oncogenic pathways. Herein, its functional duality was assessed through a thorough review of the literature and further validation in silico using The Cancer Genome Atlas (TCGA) and Gene Expression Omnibus (GEO) databases. In the literature, miR-708 was found with an oncogenic role in eight tumor types, while a suppressor tumor role was described in seven cancers. This double profile was also found in TCGA and GEO databases, with some tumor types having a high expression of miR-708 and others with low expression compared with non-tumor counterparts. The investigation of validated targets using miRBase, miRTarBase, and miRecords platforms, identified a total of 572 genes that appeared enriched for PI3K-Akt signaling, followed by cell cycle control, p53, Apellin and Hippo signaling, endocrine resistance, focal adhesion, and cell senescence regulations, which are all recognized contributors of tumoral phenotypes. Among these targets, a set of 15 genes shared by at least two platforms was identified, most of which have important roles in cancer cells that influence either tumor suppression or progression. In a clinical scenario, miR-708 has shown to be a good diagnostic and prognosis marker. However, its multitarget nature and opposing roles in diverse human tumors, aligned with insufficient experimental data and the lack of proper delivery strategies, hamper its potential as a sequence-directed therapeutic.

## 1. Introduction

MicroRNAs (miRNAs or miRs), defined as small noncoding RNAs with around 25 nucleotides that regulate gene expression in a post-transcriptional way, were initially described in 2005, and currently, over 2600 mature miRNAs have been identified in humans with characteristic expression profiles [1].

These molecules are transcribed from intergenic, intronic, or exonic DNA sequences by at least three different pathways, with the canonical pathway being the most frequent manner of their production. Transcription by RNA polymerase (RNApol) II or III results in a pri-miRNA that it is processed by the Drosha–DGCR8 complex into a pre-miRNA, which after exportation to the cell cytoplasm by the Exportin5/RanGTP, is further processed by the enzyme Dicer. The resulting double-stranded miRNA is then processed inside the RNA-induced silencing complex (RISC) into the mature form that will be able to interact with its target. The interaction between miRNAs and the target-messenger RNA (mRNA) can result in sequence-specific mRNA degradation by the action of an endonuclease present in the RISC, i.e., by decapping or deadenylation [2,3,4,5,6].

With fundamental roles in the regulation of important biological processes and normal cell functions, miRNAs are often correlated with pathologic conditions. Indeed, targeting miRNA has been explored as a therapeutic option [7]. In cancer, many miRNAs have been associated with apoptosis evasion, cell proliferation, angiogenesis, metastasis, and drug resistance [4,5,6,8,9,10], and therefore, these molecules are classified according to the function of their mRNA targets. Alterations in the miRNAs levels can activate oncogenes (oncomiRs) or inactivate tumor suppressor genes through their degradation or repression of translation, thus contributing to the initiation and progression of cancer [2]. MiRNAs also regulate angiogenesis (angiomiRs) in a cell-autonomous or non-cell-autonomous manner. The so-called pro-angiomiRs target negative regulators of signaling pathways that promote the formation of new blood vessels, while anti-angiomiRs inhibit the process by targeting positive regulators of the process [11]. Furthermore, miRNAs also contribute to tumor progression (metastamiR) by regulating several genes that favor the dissociation of neoplastic cells from the primary tumor, dissemination into body cavities, or, more commonly, the circulatory systems (lymphatic or blood vasculature), and the proliferation at ectopic sites (colonization) in response to local growth factors [12].

Nevertheless, due to their complexity, miRNAs’ mechanisms of action are associated with their location and interaction with the target mRNA, directly influencing their expression.

## 2. miR-708 Overview

In humans, miR-708 is a mirtron (together with miR-5579) generated through the splicing of intron 1 of the *ODZ4* gene [13,14], thus its processing after transcription by RNApol II does not involve Drosha (Figure 1A). *ODZ4* (located on 11q14.1) encodes the Teneurin Transmembrane Protein 4 (TENM4), a highly conserved glycosylated type II transmembrane protein with important roles during the development of the nervous system and neuronal differentiation [15]. TENM4 dysregulation has been inversely associated with relapse-free survival in breast carcinomas [15,16]. Alternatively, ovarian tumors with lower levels of this protein showed less differentiated, and more aggressive, phenotypes [17]. In prostate cancer, this gene was found inactivated through tri-methylation of histone H3 at lysine 27 (H3K27me3) but without associations to clinical features such as stage and PSA level [18]. Others have described the altered expression of TENM4 in several tumor types but with contradictory roles depending on the tissue type and little clinical significance [19].

In contrast, miR-708 is associated with different tumor types and is highlighted as a key molecule in tumor development. This miRNA was included in the mammalian microRNA expression atlas based on a small RNA library sequencing in 2007 [20], and it was initially described as dysregulated in human cervical cancer [14]. MiRBase (MI0005543) data from deep sequencing showed more abundance of the miR-708-5p strand (referred herein as miR-708) compared to miR-708-3p (also known as miR-708*) (Figure 1B). In accordance, the literature shows a more copious number of studies regarding miR-708-5p dysregulation or function, although the complemental version has also been associated with different tumor types, such as breast [21], gastric [22], and prostate [23].

Focusing on the regulatory mechanisms of *MIR708* expression, miR-708 was initially described as an endoplasmic reticulum (ER) stress-inducible miRNA, and transcriptionally induced by the transcription factor CCAAT enhancer-binding protein homologous protein (CHOP) [24]. CHOP also regulates the host *ODZ4* gene, and this co-regulation of the two transcripts induces co-expression in the brain and eyes of mice. Of note, the ER stress pathway and CHOP transcription factor are also associated with important mechanisms in cancer cells [25].

Later, it was discovered that miR-708 expression may be regulated by many different transcriptional and post-transcriptional mechanisms. For instance, its induced expression is dependent on the transcriptional activity and transactivation of the glucocorticoid receptor alpha (GRα) [26]. This regulatory pathway was firstly confirmed through luciferase reporter assays, and, in vivo, it has been shown that treatment with synthetic or natural glucocorticoids, which is a frequent conduct with palliative purposes during chemotherapy, transactivation of GRα induces miR-708 expression in ovarian and breast cancer cells and regulates metastasis via downregulation of NF-κB signaling [26].

As a mirtron, the transcriptional regulatory mechanisms for miR-708 expression are coupled with those for *ODZ4*, the host gene (they share the same promoter*). Myc and E2F1, as transcriptional activators, and CtBP2, as a co-repressor, have binding sites within the *ODZ4* promoter and may be associated with the regulation of miR-708 [13].

Epigenetic modifications are also important for miR-708 transcriptional regulation. This miRNA was found to be repressed by the polycomb repressor complex 2 (PRC2)-induced H3K27 trimethylation in metastatic breast cancer [27]. More specifically, in prostate cancer, the polycomb repressive complex subunit Enhancer of Zeste homolog 2 (EZH2) is bound to the miR-708 promoter* and represses its expression [28]. EZH2 is the catalytic subunit of PRC2 and targeting EZH2 in cancer therapy has been considered an interesting option [29].

MiR-708 is also hypermethylated in chronic lymphoblastic leukemia (CLL) cells and is inversely correlated with its expression [30]. Additionally, an enhancer region downstream of the miR-708 promoter has a distinct DNA methylation profile in this tumor, which is associated with poor outcomes [31].

In post-transcriptional regulation, the lncRNA TMC3-AS1 interacts with pre-miR-708 molecules and decreases mature miR maturation. Therefore, TMC3-AS1 overexpression increased premature miR-708 levels and decreased levels of active mature miR-708 [32]. The miRNA maturation processes may also be affected by DNA mutation. Germline mutations and somatic mutations in the RNase IIIb domain of DICER1, which is critical in the biogenesis of mature molecules, have been reported in cancer, affecting miRNA biogenesis, and are potentially associated with 3p mature microRNA strand bias. In this regard, high expression levels of miR-708-3p in the serum of a 2-year-old girl at the time of pleuropulmonary blastoma diagnosis were attributed to a germline DICER1 [33]. Mutation in other important proteins associated with miR biogenesis may also be considered as a source of miR-708 dysregulation. Additional interactions with RNA-binding proteins must be considered, although we did not find mechanisms described in the literature.

Alternatively, with important roles in many physiological processes and as a tumor driver, miR-708 also presents diverse mechanisms of action. Of course, the interaction with the 3′ untranslated region (3′UTR) of target mRNAs to induce mRNA degradation and translational repression is the better explored mechanism for miR-708. As such, this miRNA regulates different networks and many of its predicted targets have been validated. For example, it regulates cell growth and apoptosis by interacting with SMAD3 [34], Jakq [35], DKK3 [36], AKT2 [37], and caspase 2 [38]. It also regulates PI3K/AKT [39], and Wnt3a/β-catenin signaling pathways [40] and participates in epithelial-to-mesenchymal transition by targeting ZEB1, CDH2, and vimentin [21].

MiR-708 may also regulate immune evasion, by directly targeting CD47 (a transmembrane protein that inhibits phagocytosis in T cell acute lymphoblastic leukemia) and other important immunoregulatory proteins such as B7-H3 [41], KPNA4 [42], and CD38 [43]. Furthermore, miR-708 regulation of quiescence and self-renewal targeting of the focal-adhesion associated protein Tensin3 [44], further inhibits cancer metastasis and overcomes the chemoresistance [45,46].

Polymorphisms in miR binding sites at the 3’UTR of its targets may likewise affect the interaction and roles of miR-708. For example, rs1836724 (C > T), located at the 3’UTR of ErbB4, affects the interaction with the miR-708 seed region leading to higher susceptibility to breast cancer. ErbB4 and estrogen receptor 1 (ESR1) are regulated by identical miRNAs, thus the presence of the minor frequent allele weakens miR-708–ErbB4 binding and increases the availability of miR-708 to bind to ESR1 mRNA, which may be associated with a lower expression of ESR1 (ER positive phenotype) [47].

Direct targets for miR-708-3p have also been demonstrated, including disintegrin and metalloproteinase 17 (ADAM17), which depend on the GATA/STAT3 signaling pathway [48] and are associated with epithelial-to-mesenchymal transition.

Long non-coding RNAs are also miR-708 targets. For example, miR-708-5p and LINC01232 are mutually regulated in a feedback loop in HCC cells [49]. More recently, competing endogenous RNAs (ceRNAs) mechanisms, defined as crosstalk between RNA molecules sharing miRNA binding sites have been described. Such molecules sequester miRNAs in a complex regulatory network in which RNAs can regulate each other by competing for a limited pool of miRNAs [50]. In this regard, many long non-coding RNAs have been included in these ceRNA networks, including NONRATT009773.2/miR-708-5p/CXCL13 in bone cancer [51]; LINC00514/miR-708-5p/HOXB3 in cervical squamous cell carcinoma [52]; LINC00665/miR-708 and miR-142-5p/RAP1B in osteosarcoma [53]; Meg3/miR-708/SOCS3 in colorectal cancer [54]; LOXL1-AS1/miR-708-5p/USF1 and NF-κB [55,56]; and the RNAs circ_0009112 and MINCR, sponging miR-708-5p and regulating PI3K/AKT and Wnt/β-catenin pathways [57,58].

Therefore, based on this information, it is clear that the mechanisms to regulate miR-708 expression are complex, just like the complex interaction networks for its targets, justifying the different roles that these molecules may perform in different tumor types.

## 3. The Villain: MiR-708 Encouraging Tumor Growth and Progression

Since its first description, the role of miR-708 as a tumor driver, has repeatedly been described (Figure 2). Hyperexpression of this miRNA was found in colorectal cancer and with clear involvement in the fomentation of cellular growth [59,60], poorer prognosis [61], including more advanced stages of remission and relapse/metastasis, death, and shorter event free 5-year survival [62]. In vitro, anti-miR transfection increased apoptosis, reduced the number of invasive cells by negatively regulating CDKN2B expression levels [61].

Similar expression patterns have also been reported in other digestive system cancers, such as gastric carcinomas [63] and pancreatic ductal adenocarcinoma, in which the high expression of miR-708 is positively correlated with poor prognosis [64]. The biological function of this miRNA was further validated through the transfection of BxPC-3 with mimics, which rendered increased proliferation and clonogenic growth [64]. Moreover, a 3-fold upregulation of miR-708 was found to occur during the malignant transformation of intraductal papillary mucinous [65] and in pancreatic intraepithelial neoplasms, pointing to a probable contribution to the development and progression of pancreatic tumors [66].

Elevated expression levels of miR-708 have likewise been identified in laryngeal squamous cell carcinoma [67] and is a common feature in lung tumor tissues. This miRNA was described as one of the most highly over-expressed miRNAs in non-small-cell lung cancer (NSCLC) by Jang and co-workers (2012). Furthermore, in this tumor type, miR-708 was strongly associated with an increased risk of death and other clinically significant factors, such as age, sex, and tumor stage, possibly by potentiating WNT pathways [68]. Similar results were obtained by Song et al., (2019) through a systematic analysis of NSCLC datasets [68]. The analysis of potential miR-708 in this tumor showed 104 genes, from which *AKT1* and *CCND1* demonstrated higher interactions, and indicated the participation of this miRNA in cell cycle regulation. This role was functionally validated with gain-of-function experiments in vitro, and in both studies the transfection of cells with miR-708 mimics led to increased proliferation and facilitated invasion [68,69].

Moreover, miR-708 is significantly over-expressed in squamous cell lung cancer (SCC) compared to adenocarcinoma histological subtypes [70], and matched normal lung tissue samples [71]. Additionally, miR-708 was detected in sputum samples, proposing its use as a diagnostic biomarker [71].

This miRNA is differentially expressed in bladder cancer as well [72] and seems to increase under hypoxic conditions [73]. Nevertheless, its role in tumor development and maintenance remains elusive, with reports of increasing apoptosis in T24 and 5637 cells and reducing tumor growth [39] or being ineffective after its silencing [74].

Several other studies have also described miR-708 as a contributor to leukemogenesis. In childhood acute lymphoblastic leukemia (ALL) for instance, this miRNA was found in samples positive for MLL translocations at levels 4 times greater when compared with normal CD34+ progenitor cells. The difference was even more conspicuous in B-other ALL (negative for all major genetic abnormalities such as TEL/AML1 or BCR/ABL) with more than a 2000-fold increase [75]. High miR-708-5p expression was also detected by our group in childhood ALL, but compared to control bone marrow samples, expression levels varied according with the immunophenotype—higher levels were only observed in pre-B (no changes in pro-B and an opposite pattern in T-cell subtype) [76]. Moreover, significant associations of miR-708 upregulation with relapse-free survival, glucocorticoid therapy response, and disease risk stratification were also confirmed [77]. In fact, miR-708 levels are much higher than in standard and middle risk common-ALL [78].

Finally, reports of high expression of miR-708 were communicated as a part of a molecular signature of 7 miRNAs in anaplastic large-cell lymphomas, distinguishing ALK(+) forms from other peripheral T-cell lymphomas [79].

## 4. MiR-708 as a Hero: Its Role as Tumor Suppressor

As with the majority of miRNAs, the multitarget nature of miR-708 may render opposite roles depending on the cellular context (Figure 3). Thus, miR-708 has also been described as a tumor suppressor, where its loss determines the lack of post-transcriptional regulation of several oncogenes.

Downregulation of miR-708 has been revealed in glioblastoma and in renal cancer, and in both cases, its expression led to decreased proliferation rates and apoptosis in vitro [80,81]. The expression levels of miR-708 are also significantly lower in hepatocellular carcinoma (HCC) when compared to adjacent non-cancerous tissues, which are closely correlated with tumor grading and the advanced stages of metastasis. In accordance, the functional significance on migration and invasion was validated in HepG2 and SMMC-7721 cell lines in vitro [82]. Of note, miR-708 was also identified as a regulator of AKT2, which is a known player in hepatocarcinogenesis associated with unfavorable prognostic factors [38,83].

In patients with breast cancer, miR-708 expression was decreased in lymph nodes and distal metastases. In these conditions, miR-708 is transcriptionally repressed by polycomb repressor complex 2-induced H3K27 trimethylation, which results in the reduction of ERK and FAK activation, decreased cell migration, and impaired metastases [28]. Along these lines, forced expression of miR-708 in MDA-MB-231 cells decreased cell proliferation and invasion, whereas its inhibition led to contrasting results [84].

Low miR-708 expression was also associated with poor survival outcome, tumor progression, and recurrence in prostate cancer. In this tumor, its action was attributed to the lack of repression of CD44 and thereby the maintenance of stem-like cell populations [85]. In addition, miR-708 (together with miR-200a/miR-1203/miR-375) showed potential to be used as a discriminative serum biomarker for localized or disseminated tumors [86].

Moreover, expression levels of miR-708-5p are intrinsically related to osteosarcoma pathogenesis and correlated with poor prognosis and lower patient survival rates. The metastasis–suppressive role of this miRNA was demonstrated in vitro through functional assays, where its reestablishment in the osteosarcoma cell line HOS reduced migration and decreased or increased invasion rates depending on the matrix used, but without inducing any changes in cell proliferation or clonogenic capacity [87]. In this regard, validated miR-708 targets include several genes associated with adhesion control and metastasis, such as the E-cadherin regulators *ZEB2* and *BMI1* and many others [28,84,88,89,90]. More recently, Sui et al., (2019) confirmed these results, and attributed the effects of miR-708 loss to cell viability, and invasion to the consequent upregulation of URGCP (Up-regulator of cell proliferation), an oncogene that promotes the malignant behavior of cancer cells by regulating the NF-κB signaling pathway [91].

Downregulation of miR-708-5p is also seen in another pediatric bone tumor, Ewing sarcoma, and inversely associated with the presence of the EWS/FLI1 translocation. However, in this tumor type no relations with any prognostic features such as HUVOS grade, event or survival were found [92]. Nonetheless, a previous study showed that miR-708 acts as a repressor of EYA3 and might be important for chemotherapy response in this childhood tumor [93,94].

Correspondingly, miRNA-708 as a determinant of chemoresistance was described in renal cell carcinoma, in which miRNA-708 is frequently reduced. In this tumor type, its ectopic expression increased the accumulation of sub-G1 populations and cleavage of procaspase-3 and PARP, while the sensitivity to various apoptotic stimuli such as the tumor necrosis factor-related apoptosis-inducing ligand, doxorubicin (Dox), and thapsigargin in Caki cells and in xenographic models [95]. Of note, luciferase-based screening for assessing the effects of ectopic miR-708 expression in chronic lymphocytic leukemia cells (in which lower miR-708 expression result from high enhancer methylation) identified its direct interaction with IKKβ (inhibitor of kappa light polypeptide gene enhancer in B cells), thus contributing with the aggressive phenotype by allowing a greater level of activity of the NF-κB pathway [76,95,96], a master and commander of tumor drug resistance [97,98,99].

## 5. In Silico hsa-miR-708-5p Expression Analysis in Human Cancers

In order to investigate the dual role of miR-708 in human cancer, we used the data deposited in dbDEMC (https://www.biosino.org/dbDEMC/index, accessed on 05 November 2022). This database comprises expression data of multiple miRNAs obtained through large-scale investigation, by microarray or RNA sequencing (RNA-Seq). An interesting point of this database is the availability of data generated from The Cancer Genome Atlas (TCGA) and in studies deposited in Gene Expression Omnibus (GEO) from the NCBI (National Center for Biotechnology Information, Bethesda, MD, USA). Thus, we organized the data considering TCGA cohorts and GEO datasets, thus, representing two groups of analysis.

Based on the TCGA cohort’s data, we found significant differential expression patterns between tumor and normal tissues in 13 cancer types (Figure 4A). Only kidney chromophobe (KICH) and thyroid carcinoma exhibited downregulation of miR-708-5p compared to normal counterparts. The other histologies showed increased expression of miR-708-5p (Figure 4A), namely colon (COAD) and rectum (READ) cancer, both at the lower end of the digestive tract, with the highest logFC values.

In order to maintain the same interpretation, only GEO datasets from studies that compared the expression of miR-708 in tumor samples to normal samples were included. It is important to emphasize here that this analysis did not consider cancer subtypes, which can be a prominent differential factor in some tumor types. Data obtained from GEO databases can be visualized in Figure 4B. Only ovarian cancer exhibited miR-708-5p downregulation.

We then verified the predicted power of miR-708 for the 13 TCGA cohorts that showed differential expression values between tumor and normal samples by conducting a receiver operating characteristic (ROC) analysis. The results are presented in Figure 5. In fact, the greatest potential to discriminate samples were observed for COAD, READ, LUSC, and LUAD, the same samples that exhibited highest logFC values (Figure 4A). This highlights the potential use of this miRNA as a diagnostic marker for these cancer types.

### miR-708 mRNAs Targets in Human Cancers

Using the multiMiR R package version 1.2.0 [100], we investigated miR-708 experimentally validated targets by accessing data from miRTarBase, TaRbase databases, and miRecords that included validated miRNA targets, as well as an integrated sequence-based miRNA target prediction resource from 11 popular miRNA target prediction programs (DIANA, MicroInspector, miRanda, miRTarget2, miRTarget, NBmiRtar, PicTtar, PITA, RNA22, RNAhybrid, TargetSscan). Using these tools, we identified 572 miR-708–mRNAs interactions (Appendix A). Functional enrichment analysis of validated miR-708 targets through the Kyoto Encyclopedia of Genes and Genomes (KEEG) highlighted the role of these genes in PI3K/Akt signaling, which is an important proliferative pathway in cancer development, followed by cell cycle control, p53, Apellin and Hippo signaling, endocrine resistance, focal adhesion and cell senescence regulations, which are all known contributors to tumor phenotypes (Figure 6).

Further analysis of the commonalities between miR-708-predicted targets retrieved through the different databases revealed a single shared gene, namely *BMI1*; considering miRTarBase and TaRbase we found 15 common mRNAs, while between miRecords and miRTarBase, only *ZEB2* was found. Results are represented as a Venn diagram in Figure 6C.

Most of those genes are frequently dysregulated in cancer (Table 1), despite being at different levels between tumor histologies (Figure 7). Nevertheless, the subsequent analysis of expression correlations between miR-708 and its validated targets (shared in at least two databases) did not show significant relations values except for 3 genes: *BMI1*, *ZEB2* and *CNTFR*. This analysis was also performed using TCGA data, and Pearson correlations were calculated in each cohort separately. The data were organized considering each target and the correlation plot was created for TCGA cohorts that exhibited a negative Pearson correlation and a *p* value < 0.05.

For *BMI1*, the gene that was shared among the three databases, a significant negative correlation was only observed in the PRAD cohort (Figure 8A), with the same situation being observed for *ZEB2* in the LUSC group (Figure 8B). Moreover, despite the higher miR-708 expression, which was observed in tumor samples, the only gene between the 15 validated targets (shared by two databases) that exhibited the expected opposite pattern was *CNTFR* in the CHOL cohort (Figure 8C).

## 6. Experimental Evidence for mir-708 Intervention

As with many miRNAs with predictive value, miR-708 expression dysregulation may indicate specific features of different pathologies and may also potentially be used not only for intervention in personalized medicine but also as admissible biomarkers to contribute to human healthcare.

In this regard, miR-708 was firstly identified in serum from prostate cancer patients in 2014, and its inclusion in a “circulating miRNA” (c-miRNAs) panel has shown to allow the identification of disseminated versus localized tumors, evidencing its potential for staging this cancer type [92]. These so-called c-miRNAs have been repeatedly detected in many body fluids, including not only serum, but saliva, urine, sputum and even tears, which are easily collected [101] and show advantages over conventional invasive methods.

Currently, it is possible to determine specific c-miRNAs signatures that are the consequence of their active release from cells, associated to proteins, contained in extracellular vesicles such as exosomes, or in cell death bodies [101]. The c-miRNAs expression profiles can be associated with tumor size, grade, and metastasis, serving as important tools for cancer treatment. Indeed, during the last decade, different c-miRNA panels have been proposed to predict patients’ survival [102], to improve early cancer identification [103,104], and to monitor therapy response.

However, in the same way, their expression patterns vary between physiological and pathological conditions, and the profiles may also be different between samples from diverse body origins [105]. Likewise, the relation between miRNAs tumor expression and c-miRNAs is not always similar, as shown in non-small-cell lung cancer, where miR-708 is dysregulated in tumor biopsies but it is not seen on serum samples [106]. Moreover, despite their usefulness as non-invasive biomarkers for different human indolence’s [107], there are still some technical limitations that compromise their application in clinics, including the lack of reproducibility and challenges for their isolation [105].

Comparatively, as the restoration of miR-708 expression is increasingly being repeatedly explored in vitro, not only to block proliferation (vide above) but to improve chemotherapy response [23,108,109], other limitations must be overcome such as cellular uptake, bioavailability, and accumulation at the tumor site without adverse side effects [110].

Thus, improved delivery alternatives have emerged, mainly by including miRNAs within liposomes, or associating them with gold, magnetic, silica or peptide nanoformulations [111]. Along these lines, lipid nanoparticles containing a miR-708 cargo were able to induce myocardial regeneration and heart function recovery in vivo [112]. Correspondingly, in the oncology scenario, miR-708 contained within multilayered nanoparticles have efficiently decreased lung metastasis in triple-negative breast cancer orthotopic models [113], thus giving promise for future oligonucleotide-based targeted therapy.

## 7. Final Considerations

In literature, miR-708 was found dysregulated in many tumor types, including upregulation in at least eight types, showing its potential in an oncogenic role; in other seven tumor types, miR-708 was downregulated, with more tumor suppressor activity. Comparatively, this duality was also observed in TCGA cohorts’ data, where miR-708 was found differentially expressed in 13 cancer types, with downregulation in kidney chromophobe and thyroid carcinoma. Another analysis based on GEO datasets also confirmed such opposing expression patterns, with high levels in some tumor types and a low expression in others.

It is well recognized that the multitarget nature of miRNAs may render opposite roles depending on the cellular context, and this is particularly true for miR-708 based on the inconsistent direction of its dysregulation, and differentially from other pan-cancer oncomiRs [114], it is difficult to discover the common expression patterns of its validated targets across tumor types.

Considering the clinical application of miR-708 expression analysis, miR-708 dysregulation was associated with relapse/metastasis, death, and shorter event free survival. Herein, through in silico analysis we provide more evidence of miR-708 dysregulation in cancer, highlighting it as an important regulator of tumor cell networks and reinforcing its potential as a diagnostic/prognostic marker. Further studies should determine its real potential as a molecular target with direct applicability.

## Figures and Tables

**Figure 1 genes-13-02375-f001:**
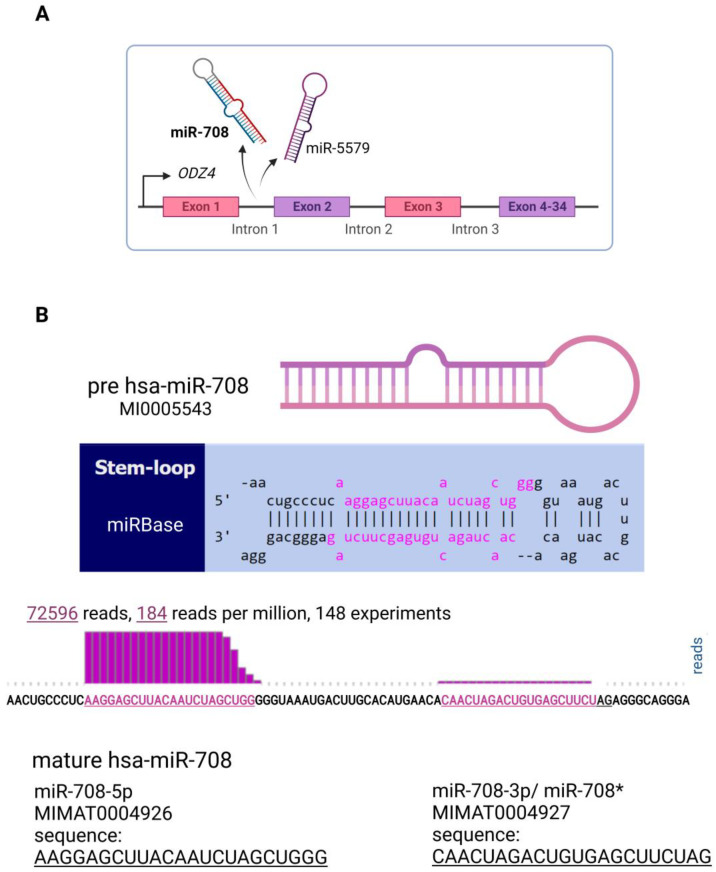
(**A**) Biogenesis of miR-708 and miR-5579 from intron 1 of the *ODZ4* gene mapped on 11q14.1. (**B**) From miRBase data (MI0005543), hsa-miR-708 stem-loop sequence and mature (708-5p and 708-3p or miR-708*) are available. The histogram represents deep sequencing diagrams with the height of bars for each nucleotide corresponding to the normalized number of nucleotides from different reads mapping to that specific nucleotide. Pink letters indicate nucleotides contained within the mature sequences. Created with BioRender.com.

**Figure 2 genes-13-02375-f002:**
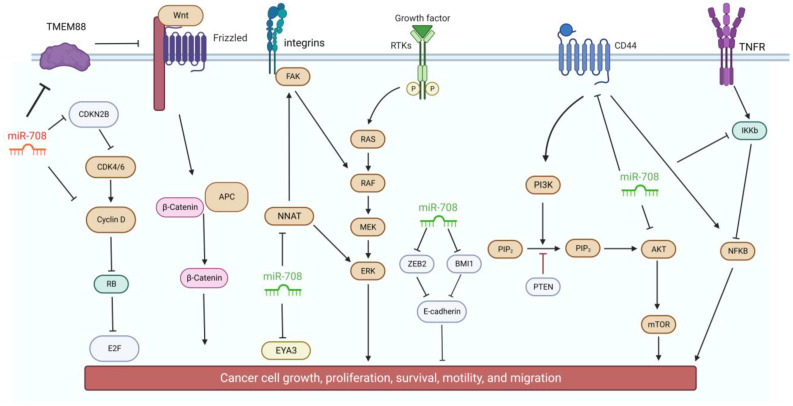
Regulatory functions of miR-708 on targets action of signaling pathways associated with cancer. The microRNA presents potential oncogenic roles when it is highly expressed (red miR-708) and tumor suppressor activity when it is downregulated (green miR-708). Created with BioRender.com.

**Figure 3 genes-13-02375-f003:**
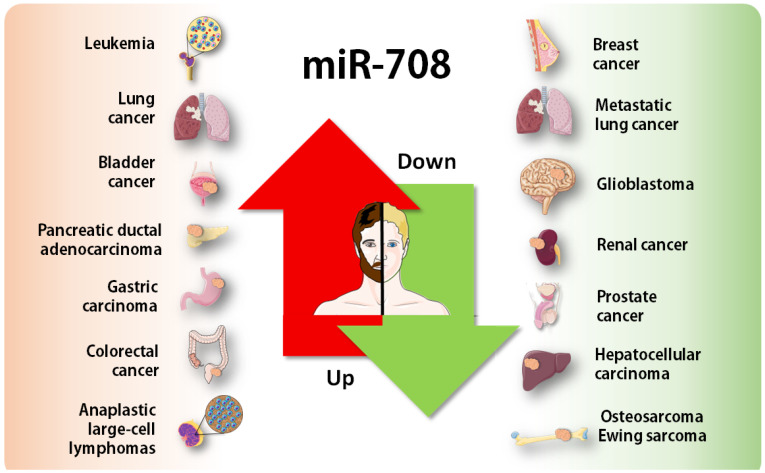
Janus-faced behavior of miR-708, acting as an oncogene (**left**) or tumor suppressor (**right**), depending on the tissue histology or cellular context. The figure was composed with the aid of illustrations from the SMART—Servier Medical Art available at https://smart.servier.com/.

**Figure 4 genes-13-02375-f004:**
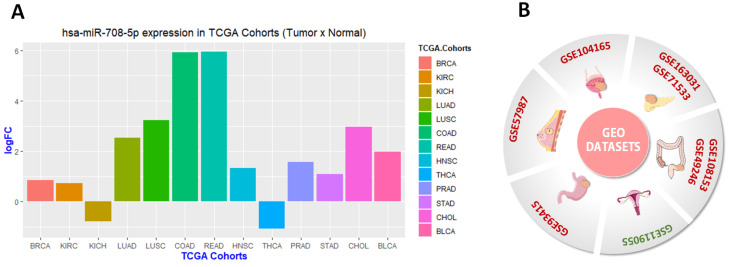
In silico analysis of miR-708 expression in human cancers. (**A**) Graphical representation of TCGA cohorts with miR-708 differential expression. The Y axis represents logFC values and the X axis TCGA cohorts. TCGA cohorts abbreviations: BLCA—bladder urothelial carcinoma; BRCA—breast invasive carcinoma; CHOL—cholangiocarcinoma; COAD—colon adenocarcinoma; HNSC—head and neck squamous cell carcinoma; KICH—kidney chromophobe; KIRC—kidney renal clear cell carcinoma; LUAD—lung adenocarcinoma; LUSC—lung squamous cell carcinoma; PRAD—prostate adenocarcinoma; READ—rectum adenocarcinoma; STAD—stomach adenocarcinoma; THCA—thyroid carcinoma. (**B**) Cancer GEO datasets that exhibited differential expression of miR-708 between tumor and normal samples. Datasets colored in red indicate miR-708 upregulation and colored in green indicate downregulation. This figure was composed with the aid of illustrations from the SMART—Servier Medical Art available at https://smart.servier.com/.

**Figure 5 genes-13-02375-f005:**
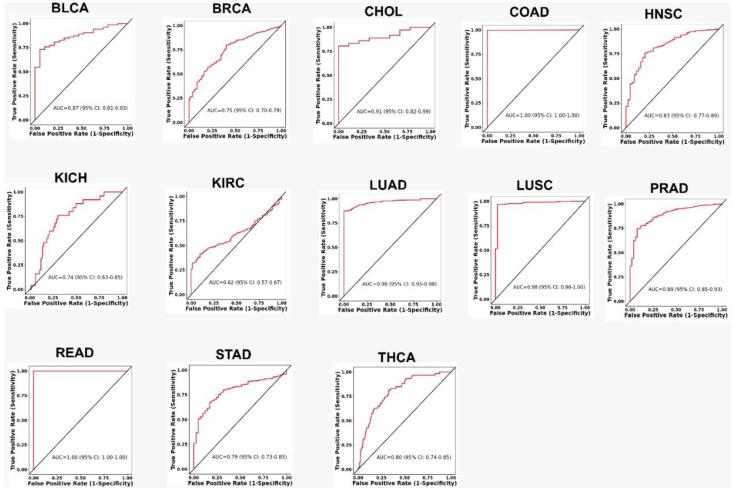
ROC curves for 13 TCGA cohorts’ data. Expression data from TCGA cohorts for tumor and normal samples were used to perform the analysis.

**Figure 6 genes-13-02375-f006:**
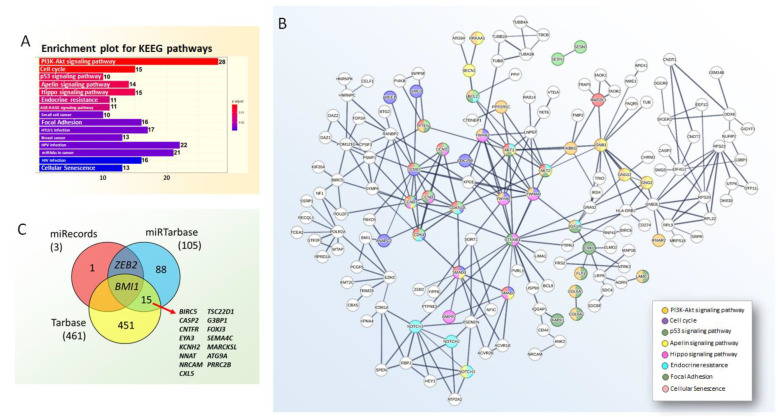
(**A**) Functional enrichment of miR-708-predicted targets through KEEG pathways. Note that most categories highlight key contributors to tumor phenotypes. (**B**) Protein–protein interactions were also accessed through the STRING v11.5 software (available at https://string-db.org/). The parameters evaluated were experiments and databases. The minimum required interaction score was 0.700, which was considered high confidence. The enrichment analysis was also performed using the same software. Network edges denote confidence and disconnected nodes were omitted. (**C**) Venn diagram representing hsa-miR-708-5p validated targets in miRecords, miRTarBase, and TaRbase database.

**Figure 7 genes-13-02375-f007:**
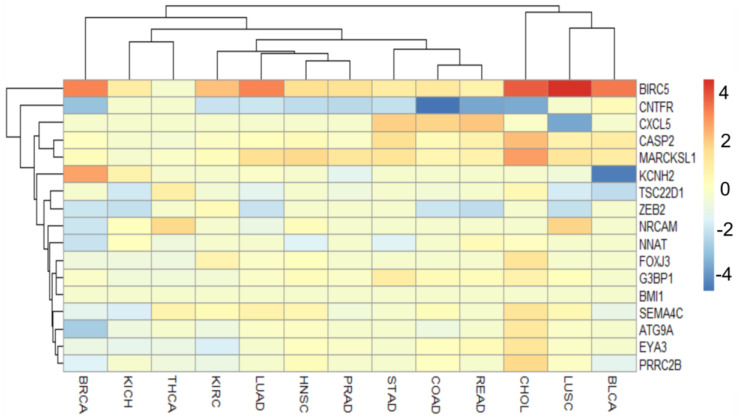
Heatmap representing miR-708-validated targets expression (shared at least between two databases) comparing tumor samples to normal samples. TCGA cohorts are represented in the columns and the individual genes in the rows. Data are represented in log2FC and genes with values equal to zero indicates a significant difference was not observed between tumor and normal samples. The heatmap was elaborated using pheatmap R package version: 1.0.12.

**Figure 8 genes-13-02375-f008:**
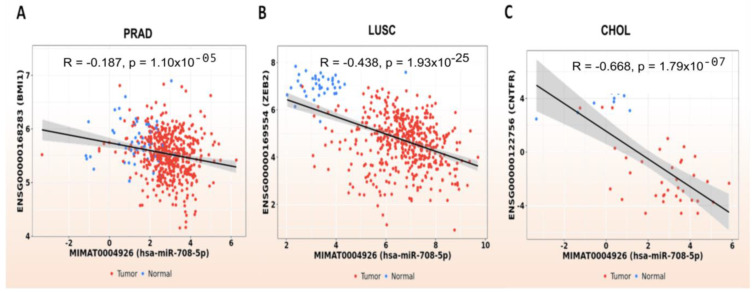
(**A**–**C**) BMI1, ZEB2 and CNTFR, and miR-708-5p Pearson´s correlation plots for PRAD, LUSC, and CHOL cohorts, respectively. The correlation plots were only generated for negative correlations with a *p* value < 0.05. For other interactions refer to Appendix A.

**Table 1 genes-13-02375-t001:** Information about the role and main mechanism in cancer cells of miR-708 validated targets expression (shared at least between two databases).

Target	Role in Cancer	Mechanism
BIRC5	oncogene	apoptosis
ATG9A	? oncogene	autophagy
CASP2	tumor suppressor	apoptosis
CNTFR	tumor suppressor	immune response
CXL5	oncogene	immune response
EYA3	oncogene	transcriptional activator
FOXJ3	? oncogene	transcriptional regulator
G3BP1	oncogene	element of Ras signal
KCNH2	oncogene	component of potassium channel
MARCKSL	oncogene	cell motility
NNAT	? tumor suppressor	component of ion channel
NRCAM	oncogene	EMT
PRRC2B	? oncogene	cell differentiation
SEMA4C	oncogene	element of MAPK cascate
TSC22D1	tumor suppressor	transcript factor

?: described with unclear results in literature. EMT: epithelial mesenchymal transition.

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
