# Peer review of "The Double Face of miR-708: A Pan-Cancer Player with Dissociative Identity Disorder"

_genes, 2022, doi:10.3390/genes13122375_

Round 1
Reviewer 1 Report
This review by Jaqueline Carvalho de Oliveira and colleagues studied A Pan-Cancer player with dissociative identity disorder, mir-708 has dual functions.
In this review, authors discussed the targets of miR-708 mRNA in human malignancies, tumor suppressor function, and tumor progression: miR-708 dysregulation in cancer, establishing it as a key regulator of tumor cell networks and boosting its potential as a diagnostic or prognostic marker
This is an interesting study, but the following concerns should be addressed during revision:
· As a reader's point of view, I would suggest that the authors include, in addition to figure 2, a figure depicting the comprehensive molecular mechanism of miR-708 as a tumor suppressor and tumor progression, along with its targets (proteins, mRNAs, and lncRNA).
· I would advise authors to include a table of miR-708-targeted coding RNAs and noncoding RNAs that are involved in tumor progression and tumor suppressor functions.
· Enhancers are critical genomic elements that can cooperate with promoters to regulate gene transcription in both normal and cancer cells. Enhancer regions can be actively transcribing and produce a type of non-coding RNA termed enhancer RNAs (eRNAs), which are a class of RNAs that do not encode proteins but are still functionally important. Rising evidence shows that eRNAs play significant roles in enhancer activation and a critical role in gene regulation, and the expression of eRNAs may be a critical factor in tumorigenesis. The essential roles of eRNAs in cancer signaling pathways are also gradually revealed, providing a new insight into cancer therapy. In addition to mRNAs and lncRNAs, I would recommend that authors discuss the functional mechanisms and importance of MiR-708 targeted eRNAs in tumor growth and progression cells, as well as their therapeutic and diagnostic value in cancer.
Author Response
“This review by Jaqueline Carvalho de Oliveira and colleagues studied A Pan-Cancer player with dissociative identity disorder, mir-708 has dual functions. In this review, authors discussed the targets of miR-708 mRNA in human malignancies, tumor suppressor function, and tumor progression: miR-708 dysregulation in cancer, establishing it as a key regulator of tumor cell networks and boosting its potential as a diagnostic or prognostic marker.
This is an interesting study, but the following concerns should be addressed during revision:
- As a reader's point of view, I would suggest that the authors include, in addition to figure 2, a figure depicting the comprehensive molecular mechanism of miR-708 as a tumor suppressor and tumor progression, along with its targets (proteins, mRNAs, and lncRNA)."
Response: We are grateful for this suggestion. For better exemplifying pathways and targets regulated by miR-708-5p in cancer cells as potential tumor suppressor or oncogenic activity, we included a new figure (Figure 2) in addition to figure showing the Janus-faced behavior of miR-708 (now figure 3).
In this new figure, we highlighted molecules targets cited in topics 3 and 4, and exemplified some potential oncogenic roles when miR are highly expressed (red miR-708) and tumor suppressor activity when down regulated (green miR-708).
- "I would advise authors to include a table of miR-708-targeted coding RNAs and noncoding RNAs that are involved in tumor progression and tumor suppressor functions".
Response: We agree that the inclusion of a table about targets is interesting. Thus, in supplementary table 1 are all experimentally miR-708 validated targets from different databases. For more details, we included in manuscript “Table 1”, including Information about involvement of these targets (shared at least between two databases) in tumor progression and suppression and main mechanism in cancer cells.
- "Enhancers are critical genomic elements that can cooperate with promoters to regulate gene transcription in both normal and cancer cells. Enhancer regions can be actively transcribing and produce a type of non-coding RNA termed enhancer RNAs (eRNAs), which are a class of RNAs that do not encode proteins but are still functionally important. Rising evidence shows that eRNAs play significant roles in enhancer activation and a critical role in gene regulation, and the expression of eRNAs may be a critical factor in tumorigenesis. The essential roles of eRNAs in cancer signaling pathways are also gradually revealed, providing a new insight into cancer therapy. In addition to mRNAs and lncRNAs, I would recommend that authors discuss the functional mechanisms and importance of MiR-708 targeted eRNAs in tumor growth and progression cells, as well as their therapeutic and diagnostic value in cancer.”
Response: eRNAs are interesting molecules bt still under explored. Trying to follow reviewer suggestion, we searched for regulatory mechanisms using two different strategies: in one of them we searched for direct linkage, via base pairing between the list of human eRNAs and hsa-miR-708-5p, however the database that contains the sequences of the most of 60,000 eRNAs was not available for download (https://hanlab.uth.edu/HeRA/m3).
The other strategy was to search for eRNAs that could activate hsa-miR-708 transcription. An important factor for our search is that miR-708 is transcribed from a TNM4 coding gene, and both are regulated by the same promoter. Therefore, we search the database: https://hanlab.uth.edu/HeRA/m3 looking for the TNM4 gene as a target. And quite interestingly, we identified the eRNA ENSR00000042698 as a potential regulator. It was not possible to search specifically for the miRNA because the database does not support it. This regulatory mechanism seems quite interesting; however, we chose not to add this topic to the manuscript because it was not possible to validate it. This mechanism drew our attention a lot, and will certainly must be the subject of future studies.
Reviewer 2 Report
The literature indicates that miR-708 has a potential oncogenic role, and it is also emphasized that miR-708 is down-regulated by tumor suppressor activity in some tumor types. These molecular mechanisms were explained comprehensively by in silico analysis and the addition of schematic diagrams added value to the manuscript. The topic is important and the authors collected comprehensive literatures in the field. I would suggest that the manuscript can be better if the authors can improve the manuscript in following aspects:
Abbreviations should be explained (use full name) at the first time and then later use abbreviations only.
The title is on the double face of Mir-708, but in the abstract the authors did not focus on the double face of Mir-708 and should emphasize its oncogene role and tumor suppressor activity.
Please be specific about these features of the Mir-708. Similarly, please in the main text be specific.
The critical roles of miR-708 in tumor cell networks should be fully discussed.
Without this information, this article would be insufficient.
Author Response
“The literature indicates that miR-708 has a potential oncogenic role, and it is also emphasized that miR-708 is down-regulated by tumor suppressor activity in some tumor types. These molecular mechanisms were explained comprehensively by in silico analysis and the addition of schematic diagrams added value to the manuscript. The topic is important and the authors collected comprehensive literatures in the field. I would suggest that the manuscript can be better if the authors can improve the manuscript in following aspects:
Abbreviations should be explained (use full name) at the first time and then later use abbreviations only.”
Response: abbreviations were explained accordingly. All changes were highlighted.
“The title is on the double face of Mir-708, but in the abstract the authors did not focus on the double face of Mir-708 and should emphasize its oncogene role and tumor suppressor activity.”
Response: the abstract was modified as suggested.
“Please be specific about these features of the Mir-708. Similarly, please in the main text be specific.
The critical roles of miR-708 in tumor cell networks should be fully discussed.
Without this information, this article would be insufficient.”
Response: We thank the reviewer for suggestions. We reinforced in abstract and manuscript the feature of miR-708 as doble effects depending on tumor types. Even more to better describe miR-708 in tumor cell networks, we included a figure (2) and a table (1). These new objects better exemplify pathways and targets regulated by miR-708-5p in cancer cells as potential tumor suppressor or oncogene.
However, we did not include a lot of new this information about tumor cell networks in text because this could make the manuscript tedious, and our focus was to describe miR-708 deregulation in tumor types (including literature search and public dataset analysis), giving some information about cell networks with references. In this manner, more details about networks may be accessed by interest.